# OpenReview forum: "BackdoorAlign: Mitigating Fine-tuning based Jailbreak Attack with Backdoor Enhanced Safety Alignment"
_NeurIPS.cc/2024/Conference — NeurIPS 2024 poster_

### Official Review · Reviewer_bX8S · 2024-07-08

**Soundness:** 4
**Presentation:** 4
**Contribution:** 4
**Rating:** 8
**Confidence:** 5

**Summary:**

This paper introduces a defence against jailbreak fine-tuning attacks that markedly improves over the baseline suggested by Qi et al. Their method works by implanting a safety backdoor that is subsequently used during inference and show that it is effective on preventing few shot fine-tuning attacks across a variety of controlled settings.

**Strengths:**

Generally this paper is very well written, novel, and an extremely valuable contribution to the emerging threat of fine-tuning attacks. Not only do they convincingly demonstrate their defence is effective on the settings presented by Qi et al, and that their method indeed improves over a set of controls for different settings like using a natural language secret prompt, not using category-wise safety samples and in a setting of mixed safe and unsafe samples. I would be excited to see this paper accepted.

**Weaknesses:**

There are a few clarity issues in the paper but the main issues I’d like to see addressed are:

(1) The “pure_bad” dataset construction details are insuffecient. What is meant by redteaming?(181) What process was used for it? How was this dataset constructed? What format?  Where are the examples? Why wasn’t Qi et al. or other already existing harmful sample datasets used? Section A.2 is not enough details. Without these details we cannot trust as a reader that these were actually harmful samples.

(2) I think that the authors fail to discuss Prompt Injection attacks that leak the secret prompt. In a LLMaaS FJAttack threat model, what if the attacker gets access to the secret prompt? As mentioned in the paper and the motivation for chosing a non-semantically meaninfdul secret prompt, this means the attacker could then finetune against this prompt. I would recommend the authors at least discuss this limitation but would encourage them to add an experiment showing the effectiveness of this adaptive attack and how likely standard prompt injection methods for prompt leakage are liable to work.

(3) This is minor and is related to a weakness below but 10% of safety samples needed seems like a high cost for a defence (for example this is 10k samples for a 100k training set). Perhaps though this won’t be the case for much stronger attacks. We just don’t know what is required for stronger attacks based on the results in this paper.

This final weakness is more of what I would have liked to see to increase my score higher and what I think would improve the paper but is likely too much to ask to see addressed during the review period:

While the precedant of 100 sample attacks is set in Qi et al. I don’t think that the attack strength is high enough to be able to truly assess this method, realistically it seems like users might use thousads or more samples. I would encourage the authors to at least devise a 1k and 10k setting from a dataset like BeaverTails as is being done in other contemporaneous works. This important because without it: we are not sure that the 10% safety example mix holds, we are not sure how this method operates on more realistic attacks (for example right now there exist 10k+ harmful sample datasets on huggingface - would this defend against those? if not we should know as a communtiy so we can develop stronger defences, if so this defence is potentially promising). I’d also like to have seen more settings from Qi et al so for example varying learning rate, epoch numbers, and smaller attack sizes.

**Questions:**

### Suggestions

I think that this work could benefit from existing defences gainst FJAttacks. Certianly “Henderson, P., Mitchell, E., Manning, C., Jurafsky, D., & Finn, C. (2023, August). Self-destructing models: Increasing the costs of harmful dual uses of foundation models. In *Proceedings of the 2023 AAAI/ACM Conference on AI, Ethics, and Society* (pp. 287-296).” should be discussed (of course its not a defence against this exact threat model but it has a very similar motivation, threat model, problem setting etc.)

Other work that is contemporaneous so is optional to add but would nonetheless enhance discussion in the paper to compare and constrast these methods:

- Zhou, X., Lu, Y., Ma, R., Gui, T., Zhang, Q., & Huang, X. (2023). Making harmful behaviors unlearnable for large language models.
- Huang, T., Hu, S., & Liu, L. (2024). Vaccine: Perturbation-aware alignment for large language model
- Rosati, D., Wehner, J., Williams, K., Bartoszcze, Ł., Atanasov, D., Gonzales, R., ... & Rudzicz, F. (2024). Representation noising effectively prevents harmful fine-tuning on LLMs.

One thing that shakes out of at least Rosati et al (in particular their earlier work before RepNoise) and Henderson et al that you do not address is the limitation of your setting to only LMaaS settings. What about settings where the attacker has complete access to the model? Even in the LMaaS case there is the risk of stealing the weights so it would be good to consider whether you have any thoughts here for discussion.

Another citation worth discussing is Korbak, T., Shi, K., Chen, A., Bhalerao, R. V., Buckley, C., Phang, J., ... & Perez, E. (2023, July). Pretraining language models with human preferences. In International Conference on Machine Learning (pp. 17506-17533). PMLR. Although it uses pre-training with safety tokens, the approach has an interesting parallel.

### Notes and suggestions

2: Requires is probably a better word than “request” here

93: Calude 2

98-99,117: Missing spaces before citations

123: I don’t think its correct to say “widely used” since FJAttacks really only consist of 5 or 6 papers at this point.

126: I think exposure is a little bit too vague since in-context learning attacks could also be compromising through exposure

Equation 1 and 3: A very small optional nit pick but I think we usually represent this as an expectation of the negative log loss over the dataset distribution since the actual computation isn’t a sum or losses but a mean. But I see how this formulation has an advantage for conciseness.

136: It would be useful to cite Qi et al. again here and perhaps “Zhan, Q., Fang, R., Bindu, R., Gupta, A., Hashimoto, T., & Kang, D. (2023). Removing rlhf protections in gpt-4 via fine-tuning.” which was published at NAACL this year.

149: I feel like we should reserve this double line notation for asymmetric divergence or parallel lines and use a more standard contenation operator like “+”. I see what you might be trying to do with s_i conditioned on secret prompt s but its clearer to use a standard concatenation operator.
147,149+equation 2: I would find it clearer and more correct if you used a different notation for the secret prompt since now “s” is overloaded to mean both the whole system prompt for the index i and only the secret prompt so its confusing.

178: Please add the version of GPT-3.5 for replicability

192: 1 times what learning rate?

238-239: I don’t agree with this finding in general - from Table 1 - it seems like all you can say is a decline in utility ARC-challenge.

267-274: Can you say more about why you think this is the case?

290-293: Insightful!

295-301: Can you provide the hyperparameters used for LoTA here or in an appendix?

345: I don’t agree that 10% is a very small set of safety examples.

489: generate

**Limitations:**

Aside from what I mentioned above they discuss the limitations very well.

---

> ### Author Rebuttal · Authors · 2024-08-07
>
> > Question A: “pure_bad” dataset construction details.
>
> The “pure_bad” dataset used in our experiments consists of 100 harmful question-answer pairs. These pairs are exactly the same as the harmful samples used in Qi et al.'s work, which were subsampled from the Anthropic red team dataset [1]. In this context, red teaming refers to the use of jailbreak attacks to collect harmful examples, aimed at evaluating the model’s robustness. As detailed in Appendix A.2, the data is formatted according to the OpenAI Standard Data Format. This includes using the same system prompt, with safety-related questions labeled as “USER INPUT” and harmful answers as “MODEL OUTPUT.”
>
>
> [1] Ganguli, Deep, Liane Lovitt, Jackson Kernion, Amanda Askell, Yuntao Bai, Saurav Kadavath, Ben Mann et al. Red teaming language models to reduce harms: Methods, scaling behaviors, and lessons learned. arXiv preprint arXiv:2209.07858 (2022).
>
> > Question B: The leakage of the secret prompt through prompt injection attacks.
>
> To determine if a prompt injection attack could leak our secret prompt, we test the five attack queries presented in the paper [1] to assess their ability to extract the system prompt from the fine-tuned Llama-2-7B-Chat model. Surprisingly, the fine-tuned model identified all these prompt extraction attempts as harmful behavior and refused to disclose its system prompt. This resistance to leakage is partly due to the secret prompt's inherent capability to enhance safety by protecting personal privacy. Additionally, this further demonstrates that the secret prompt is not easily extracted by malicious users with adaptive attacks.
>
> [1] Zhang, Yiming, Nicholas Carlini, and Daphne Ippolito. Effective prompt extraction from language models. arXiv preprint arXiv:2307.06865 (2024).
>
> > Question C: Defense with a larger training set and stronger attack with more harmful examples.
>
> We utilize 10% safety examples in the initial setting due to the involvement of harmful examples only. In a more realistic scenario, the dataset provided by users predominantly consists of benign examples for specific applications, along with a few undesirable harmful examples. As detailed in Section 5, we incorporated the safety examples into a fine-tuning dataset comprising 1,000 training samples. This represents only about 1% safety examples of the total training set yet still achieves satisfactory defense performance. Here we also report the Average Epoch Training Time under 1xNVIDIA A100 GPU to calculate the exact extra time introduced by the 1% safety examples. From the table, we can observe that the 1% extra safety examples only bring about 1s extra GPU time for our defense.
>
> |Defense Method| Avg Epoch Training Time of Llama-2-7B-Chat |
> | ----| ----  |
> |No Defense | 159.24s   |
> |Ours |  160.44s |
>
> To determine if our method can defend against stronger attacks involving thousands of harmful examples, we evaluated our backdoor enhanced safety alignment under fine-tuning with 1,000 harmful examples randomly sampled from BeaverTails under Llama-2-7B-Chat with 11 safety examples, repeating 10 times. The repeating strategy here is to make sure the model can sample enough times of safety examples without collecting new safety question-answer pairs. The results, displayed in the following table, indicate that our method can reach better defense performance under stronger attacks compared with the no-defense setting.
>
> | Defense Method | 100 harmful examples | 1000 harmful examples  |
> | ----  |----  |----  |
> | No Defense | 94.91  | 88.00 |
> | Ours | 3.64  | 3.64 |
>
>
> > Question D: Suggestions and notes
>
> Thank you very much for your suggestions and feedback! We will incorporate more discussions on your suggested papers in the final version. Additionally, we will continue refining our paper to correct the typos and clarify the descriptions you pointed out.
>
> Reply to questions in the notes:
>
> 178: Our GPT-3.5 version is gpt-3.5-turbo-0613.
>
> 192: Times of learning rate multiplier is one parameter used in OpenAI Fine-tuning API. The exact learning rate used in the API is protected by OpenAI and unknown to us.
>
> 238-239: Here, we observe a decline in the MT-Bench Score of Llama 2 and the ARC Challenge Accuracy of GPT-3.5 when comparing performance before and after the FJAttack (between the No Attack and No Defense settings). These declines were already present before implementing the defense, demonstrating that they were not introduced by our subsequent defense.
>
> 267-274: The choice of secret prompt length is primarily based on our empirical findings. Figure 4 clearly illustrates that a longer secret prompt helps reduce the attack success rate. However, considering that longer prompts for LLMs incur additional inference costs, we have chosen 150 as the final length. This selection considers a balance between effectiveness and efficiency.
>
> 295-301: For our LoRA fine-tuning, we use $lora\\_alpha=16$, $lora\\_dropout=0.1$ and the lora attention dimension $r=8$ with other hyperparameters as default ones.

---

> ### Comment · Reviewer_bX8S · 2024-08-11
> **Thanks again to the authors**
>
> I appreciate the authors efforts in responding and hope that the excersize was helpful in clarifying the paper for final revision.
>
> As stated previously, I believe this is a novel and significant contribution and I hope to see it accepted.

---

> > ### Author Response · Authors · 2024-08-11
> >
> > Thanks again for acknowledging our work. We will make the corresponding revisions in our final version.

---

### Official Review · Reviewer_oEoJ · 2024-07-10

**Soundness:** 3
**Presentation:** 3
**Contribution:** 3
**Rating:** 6
**Confidence:** 4

**Summary:**

The authors introduce the Backdoor Enhanced Safety Alignment method, which uses prefixed safety examples with a secret prompt acting as a backdoor trigger to ensure safety responses during inference. This approach aims to maintain the safety alignment of LLMs with minimal safety examples and without compromising their benign performance.

**Strengths:**

1. The method requires only a small number of prefixed safety examples to achieve significant improvements in safety performance.

2. The paper conducts extensive experiments, including ablation studies on token length, safety samples and real-world scenarios, to validate their approach.

**Weaknesses:**

1. The paper only uses PolicyOriented Safety Evaluation Benchmarks for harmlessness evaluation, which may not fully capture the method's impact on overall model performance in diverse scenarios.

2. This method still requires a very small set of safety examples for fine-tuning, which is only applicable to the settings of Language-Model-as-a-Service.

3. Since determining refusal answers based on a list of rejection keywords is highly inaccurate, some open-sourced Judge models, such as LlamaGuard2 [1] or MD-Judge [2] can be utilized for attack success rate evaluation.

[1] Inan, Hakan, et al. "Llama guard: Llm-based input-output safeguard for human-ai conversations." arXiv preprint arXiv:2312.06674 (2023).

[2] Li, Lijun, et al. "Salad-bench: A hierarchical and comprehensive safety benchmark for large language models." arXiv preprint arXiv:2402.05044 (2024).

**Questions:**

See weakness part.

---

> ### Author Rebuttal · Authors · 2024-08-07
>
> > Question A: Various benchmarks for harmlessness evaluation.
>
> To demonstrate the effectiveness of our method across different scenarios, we also apply the AdvBench [1] and HarmBench [2] benchmarks, which are widely used to assess robustness against jailbreak attacks, to evaluate safety alignment performance. The Attack Success Rates (ASR) for different defense methods under the Llama-2-7B-Chat model are presented in the table below.
>
> | Defense Method | PolicyOriented Safety Evaluation Benchmark | AdvBench | HarmBench
>  | ----  |----  |----  |----  |
> | No Attack | 3.27  | 0.00 | 11.25 |
> | No Defense | 94.91  | 96.54 | 96.25 |
> | Baseline |  34.91 | 40.19 | 73.12 |
> | Ours | 3.64  | 0.00 | 3.75 |
>
> The table shows that our method significantly outperforms the baseline defense across all three benchmarks, demonstrating the effectiveness of our approach.
>
> [1] Zou, Andy, Zifan Wang, J. Zico Kolter, and Matt Fredrikson. Universal and transferable adversarial attacks on aligned language models. arXiv preprint arXiv:2307.15043 (2023).
>
> [2] Mazeika, Mantas, Long Phan, Xuwang Yin, Andy Zou, Zifan Wang, Norman Mu, Elham Sakhaee et al. Harmbench: A standardized evaluation framework for automated red teaming and robust refusal. arXiv preprint arXiv:2402.04249 (2024).
>
> > Question B: Only applicable to the settings of Language-Model-as-a-Service
>
> First, we want to emphasize that the Language-Model-as-a-Service (LMaaS) has been widely used by companies in practice (e.g., OpenAI). We believe this is a highly practical and commonly used setting, utilized by a significant number of users. Thus, providing an effective and efficient defense method against fine-tuning based jailbreak attacks in this setting represents significant progress in this field.
>
> Here, we still clarify that our method still works on the open-source models under the threat model, where the attacker can upload the data to perform the fine-tuning-based jailbreak attack but can not control the inference template. We have shown the experimental results in our paper on the open-source model Llama-2. We also add experiments on mistral-7B. The results are as follows. It shows the effectiveness of our method on other open-source models. (Ours is 22.55 while the initial model ASR is 17.09. We are significantly better than the baseline method.)
>
> | Defense Method | Llama-2-7B-Chat | Mistral-7B-Instruct-v0.2|
>  | ----  |----   |----  |
> | No Attack | 3.27  | 17.09  |
> | No Defense | 94.91  | 97.09  |
> | Baseline |  34.91  | 44.00  |
> | Ours | 3.64  | 22.55 |
>
>
>
> > Question C: Open-sourced Judge models for ASR evaluation.
>
> The ASR computed using rejection keywords is a simple and efficient evaluation method employed in our experiments. Additionally, we include the Harmfulness Score, which uses GPT-4 as the judge model. Our results demonstrate that evaluations using both rejection keywords ASR and the Harmfulness Score consistently support our conclusions.
> However, evaluating with GPT-4 incurs significant costs due to API usage. To provide a more accurate and cost-saving evaluation method, we utilize the open-source models LlamaGuard2 and MD-Judge for safety classification. These models compute the ASR by calculating the proportion of ‘unsafe’ labels generated under the Llama-2-7B-Chat model. The results are presented in the following table.
>
> | Defense Method | ASR | Harmfulness Score | LlamaGuard2 ASR | MD-Judge ASR |
> | ----  |----  |----  |----  |----  |
> | No Attack| 3.27  |1.11  |  0.00 | 8.73 |
> | No Defense | 94.91  | 4.68 | 64.36 | 91.27 |
> | Baseline |  34.91 | 2.49 | 24.73 | 40.00 |
> | Ours | 3.64  | 1.22 | 3.64 | 8.00 |
>
> The experimental results in the table show that our defense method significantly outperforms the baseline in effectively defending against fine-tuning-based jailbreak attacks across all harmlessness evaluation metrics, including assessments using the open-source judge models LlamaGuard2 and MD-Judge.

---

> > ### Author Response · Authors · 2024-08-11
> > **Look forward to your reply**
> >
> > Dear Reviewer oEoJ,
> >
> > The deadline for the discussion period is approaching. We have provided our rebuttal material and hopefully could address your concerns. Your feedback is highly valuable to us, and we would greatly appreciate it if you could take some time to review our response.
> >
> > Best Regards,
> >
> > Authors

---

> ### Comment · Reviewer_oEoJ · 2024-08-13
>
> Thanks for the response. As the response solves my concerns, I will increase my score. It would be great to see those added contents in the revised version.

---

### Official Review · Reviewer_D4Wj · 2024-07-11

**Soundness:** 3
**Presentation:** 3
**Contribution:** 3
**Rating:** 5
**Confidence:** 4

**Summary:**

In this paper, the authors present a new approach to defending LLMs against the fine-tuning-based Jailbreak Attack (FJAttack). The FJAttack exploits the fine-tuning process by introducing harmful examples into the dataset, compromising the model's safety alignment. The proposed method, Backdoor Enhanced Safety Alignment, uses a backdoor trigger mechanism to incorporate safety examples into the fine-tuning process. By prefacing safety examples with a secret prompt (the backdoor trigger), the model learns to associate this prompt with safe responses. The secret prompt is then prepended to user inputs during inference, ensuring safe outputs even when the model is exposed to harmful queries. The paper demonstrates the effectiveness of this method through extensive experiments, showing significant improvements in safety without compromising the model's performance on benign tasks.

**Strengths:**

1. The introduction of a backdoor mechanism for safety alignment is innovative and provides a new perspective on defending LLMs against fine-tuning attacks.

2. The method requires only a small number of safety examples to be effective, addressing the inefficiency of previous approaches that needed large datasets.

3. The paper provides a thorough analysis, including ablation studies and comparisons with baseline methods, to validate the robustness of the proposed method.

**Weaknesses:**

1. The method heavily relies on a secret prompt, which poses a security risk if the prompt is discovered or guessed by malicious users. Additionally, the algorithm for generating the secret prompt is overly simplistic, relying on random generation. Consequently, the improvement it offers is not significant in terms of both defense and utility.

2. Despite being small, the need for fine-tuning with safety examples introduces an extra cost, which may not be feasible in all settings.

3. The scalability of the method to larger models or more complex tasks has not been extensively explored.

**Questions:**

1. How is the secret prompt designed to ensure it does not interfere with the semantic meaning of benign queries?

2. How scalable is the proposed method when applied to LLMs with varying architectures and sizes?

**Limitations:**

1. Although the method is efficient, it still requires a small set of safety examples, which may not always be readily available. Additionally, the efficiency is not well demonstrated, as there is no clear comparison to baseline methods to show the extent of improvement.

2. As the method uses a secret prompt, there is a risk of adaptive attacks where attackers design strategies to circumvent the backdoor mechanism.

---

> ### Author Rebuttal · Authors · 2024-08-07
>
> > Question A: Concerns about the security risks of the secret prompt; improvement is not significant
>
> Our defense method is primarily designed for the Language-Model-as-a-Service (LMaaS) based threat model, where attackers are only permitted to upload a fine-tuning dataset to perform the fine-tuning based jailbreak attacks, while the processes of fine-tuning and inference remain under the control of the LLM service providers. In this setting, the secret prompt is created by and known only to the model provider, making it difficult for malicious users to discover or guess.
>
> To further enhance the stealthiness of the secret prompt, we employ dynamically generated random tokens, which prevents attackers from easily guessing the secret prompts. This is a key design of our method. The results shown in Table 4 further highlight the effectiveness of selecting random tokens as the secret prompt, in comparison to other methods like the Default or GPT-4 Generated secret prompts with specific semantic meanings.
>
> We disagree with the reviewer’s comment that our performance is not significant in terms of both defense and utility.  We conduct comprehensive experiments to evaluate the defense and utility of our method. The results show our method **significantly improves safety alignment without compromising model utility** . As shown by the results in Table 1, our method achieves a 2.82 lower Harmfulness Score and a 45.09% reduction in ASR, while maintaining comparable ARC-Challenge and MMLU accuracy and even achieving a slightly higher MT-Bench score compared to the baseline under GPT-3.5.
>
> > Question B: Extra cost of our method; comparison to baseline methods;  not feasible in all settings.
>
> Thank you for your questions and suggestions. Despite our method will introduce extra costs, we do not think the extra cost is high since our method requires only 11 additional safety examples. To more accurately assess the additional costs associated with these safety examples, we calculated the Average Epoch Training Time for the Llama-2-7B-Chat using a single NVIDIA A100 80GB GPU. The details of these extra costs are presented in the following table:
>
> |Defense Method| Avg Epoch Training Time of Llama-2-7B-Chat |
> | ----| ----  |
> |No Defense | 16.40s  |
> |Ours |  18.77s  |
>
> From the table, it is evident that compared to the No Defense setting, our method requires only an additional 2 seconds of GPU time to defend against fine-tuning based jailbreak attacks. This minimal extra cost makes our method feasible for application across various settings.
>
> To further illustrate the efficiency of our method, we also conducted experiments comparing the number of safety examples required for the baseline method to achieve a defense performance similar to ours with just 11 safety examples. These experiments were performed using the Llama-2-7B-Chat model, and the results of the attack success rate are detailed in the table below.
>
> |Defense Method| Number of Safety Examples | ASR |
> | ----| ----  |----  |
> |Baseline  | 11 | 34.91  |
> |Baseline  | 100 |  33.09  |
> |Baseline  | 200 |  9.82  |
> |Baseline  | 300 |  4.73  |
> |Ours |   11 |  3.64  |
>
> The results in the table above indicate that to achieve a safety performance comparable to our method, the baseline defense approach requires 300 safety examples, more than 27 times the 11 safety examples. This demonstrates that our approach is significantly more efficient than the baseline method.
>
> Here we hope to highlight that our safety examples are constructed from various harmful categories (policies) instead of data instances. If we can know the harmful categories (policies), it is easier to construct safety examples. Thus, we believe our method is feasible. To evaluate the transferability of our constructed safety examples, we evaluate the model trained by our constructed safety examples among other safety evaluation benchmarks. The Attack Success Rates (ASR) for different defense methods under the Llama-2-7B-Chat model are presented in the table below.
>
> | Defense Method | PolicyOriented Safety Evaluation Benchmark | AdvBench | HarmBench
>  | ----  |----  |----  |----  |
> | No Attack | 3.27  | 0.00 | 11.25 |
> | No Defense | 94.91  | 96.54 | 96.25 |
> | Baseline |  34.91 | 40.19 | 73.12 |
> | Ours | 3.64  | 0.00 | 3.75 |
>
> The table shows that our method is still effective among other benchmarks and can significantly outperform the baseline defense across all three benchmarks with a large margin.
>
> > Question C: Scalability of our method on various architectures.
>
> In our paper, we have included two different LLMs with different architectures and sizes: Llama-2-7B-Chat and GPT-3.5. To better assess the scalability of our defense method for different architectures, we further conduct experiments using the Mistral-7B-Instruct-v0.2 model. The defense performance is evaluated by presenting the keyword list attack success rates under various defense methods, as detailed below.
>
> | Defense Method | Llama-2-7B-Chat | Mistral-7B-Instruct-v0.2|
>  | ----  |----  |----  |
> | No Attack | 3.27   | 17.09  |
> | No Defense | 94.91   | 97.09  |
> | Baseline |  34.91  | 44.00  |
> | Ours | 3.64   | 22.55 |
>
> The results in the table reveal that our defense method outperforms the baseline across various architectures, demonstrating the generalizability of our approach to different LLMs.
>
> > Question D: Scalability of our method for more complex tasks.
>
> Our paper addresses not only the direct attack setting, where only harmful data is used for the attack but also a more complex task where harmful examples are mixed into a fine-tuning dataset. We evaluate the defense effectiveness in two practical fine-tuning tasks: dialog summary and SQL generation, both with harmful examples included in the fine-tuning dataset. The results in Table 6 show that our method outperforms the baseline defense approach without compromising fine-tuning performance, demonstrating the scalability of our method for more complex tasks.

---

> ### Author Response · Authors · 2024-08-07
> **Additional Part of the Rebuttal**
>
> > Question E: Secret prompt design to maintain the semantic meaning of benign queries
>
> Thank you for your question.  In our paper, we have evaluated the performance on various widely-used benchmarks, including the ARC Challenge, MMLU, and MT-Bench. The results are shown in Table 1 in the paper. It empirically demonstrates that backdoor triggers do not significantly harm natural generation outputs. For instance, under  Llama-2-7B-Chat, the model fine-tuned with our defense can achieve the best ARC Challenge Acc, but slightly lower performance in MMLU Acc and MT-Bench Score.
>
> The potential reasons are as follows:  despite our method inspired by the traditional backdoor attack to build a correlation between the trigger and safety response, we hope to highlight that our method works in a totally different setting compared with traditional backdoor attacks.  **Fine-tuning jailbreak attacks** focus on the fine-tuning stage. At this stage, the initial model has already been trained on a very large corpus, which endows it with strong generation performance (mapping benign questions to normal responses) and robust safety alignment (mapping harmful questions to refusal responses). It’s important to note that before this stage, the model has NEVER learned the ability to map benign data to refusal responses.
>
> Within this context, what our method does is to strengthen the mapping from the trigger + harmful question and safety response with a trigger, while still maintaining the model’s initial generation performance, by using a small amount of triggered data.  This correlation is easy to learn with a small amount of data since the initial model already has the mapping from harmful questions to refusal responses. However, such a trigger is hard to generalize to benign question + trigger to refusal response since the mapping from benign data to refusal response does NOT exist in the initial model. The small amount of triggered data is not enough to build a correlation between the trigger + benign question and refusal response. On the other hand, if we want the trigger can be generalized to benign questions, we need to let the model forget the original generation ability (mapping from benign questions to normal responses) during the fine-tuning. In this way, the model’s initial generation performance will also significantly drop, which is not aligned with the principle of fine-tuning.
>
> > Question F: Risk of adaptive attacks.
>
> To conduct adaptive attacks, attackers should first determine the secret prompt used in our method, which can be achieved through prompt injection attacks aimed at leaking the system prompt. Thus, we test the five prompt injection attack queries presented in paper [1] to assess whether they can extract the secret prompt from the fine-tuned Llama-2-7B-Chat model. Surprisingly, the fine-tuned model identified all these prompt extraction attempts as harmful behavior and refused to disclose its system prompt. This resistance to leakage is partly due to the secret prompt's inherent capability to enhance safety by protecting personal privacy. Additionally, this further demonstrates that the secret prompt is not easily extracted by malicious users with adaptive attacks.
>
> [1] Zhang, Yiming, Nicholas Carlini, and Daphne Ippolito. Effective prompt extraction from language models. arXiv preprint arXiv:2307.06865 (2024).

---

> > ### Author Response · Authors · 2024-08-11
> > **Look forward to your reply**
> >
> > Dear Reviewer D4Wj,
> >
> > The deadline for the discussion period is approaching. We have provided our rebuttal material and hopefully could address your concerns. Your feedback is highly valuable to us, and we would greatly appreciate it if you could take some time to review our response.
> >
> > Best Regards,
> >
> > Authors

---

### Official Review · Reviewer_uBHD · 2024-07-13

**Soundness:** 3
**Presentation:** 3
**Contribution:** 3
**Rating:** 6
**Confidence:** 4

**Summary:**

This paper proposes a defense method against fine-tuning-based jailbreaking attacks on close-source LLM services. The main insight is to add a backdoor trigger to safe prompts incorporated during the fine-tuning, and use the trigger as a prefix during inference.

**Strengths:**

1. This paper focuses on a trendy and important AI safety problem.
2. The evaluation considers diverse settings, including both malicious fine-tuning and simple task-specific fine-tuning.
3. The ablation study covers various components of the proposed method.

**Weaknesses:**

1. The reason why the backdoor triggers are not harmful to natural generation may be further explained or empirically studied. For general backdoor machine learning, the trigger is to break the performance of the model when injected. How can the safe triggers not affect the LLM’s performance?
2. The defense uses a system prompt during inference to improve the generation safety. Therefore, some prompt-based defenses may need to be compared as baselines, like self-reminder [1] and In-context defense [2].
3. The method cannot defend against fine-tuning attacks on open-source models, which should be acknowledged as a limitation and specified in the title (e.g., Mitigating Fine-tuning based Jailbreak Attack on cloud services …).

[1] Defending ChatGPT against jailbreak attack via self-reminders

[2] jailbreak and guard aligned language models with only few in-context demonstrations

**Questions:**

See weaknesses.

**Limitations:**

See weaknesses.

---

> ### Author Rebuttal · Authors · 2024-08-07
>
> > Question A: Concerns about the safe triggers affecting the LLM’s performance.
>
> Thank you for your question. In our paper, we have evaluated the performance on various widely-used benchmarks, including the ARC Challenge, MMLU, and MT-Bench. The results are shown in Table 1 of our paper. It empirically demonstrates that backdoor triggers do not significantly harm natural generation outputs. For instance, under  Llama-2-7B-Chat, the model fine-tuned with our defense can achieve the best ARC Challenge Acc, but slightly lower performance in MMLU Acc and MT-Bench Score.
>
> The potential reasons are as follows: despite our method inspired by the backdoor attack to build a correlation between the trigger and safety response, we hope to highlight that our method works in a totally different setting.  **Fine-tuning jailbreak attacks** focus on the fine-tuning stage. At this stage, the initial model has already been trained on a very large corpus, which endows it with strong generation performance (mapping benign questions to normal responses) and robust safety alignment (mapping harmful questions to refusal responses). It’s important to note that before this stage, the model has NEVER learned the ability to map benign data to refusal responses.
>
> Within this context, what our method does is to strengthen the mapping from the trigger + harmful question and safety response, while still maintaining the model’s initial generation performance, by using a small amount of triggered data. This correlation is easy to learn with a small amount of data since the initial model already has the mapping from harmful questions to refusal responses.
> However, such a trigger is hard to generalize to benign question + trigger to refusal response since the mapping from benign data to refusal response does NOT exist in the initial model. The small amount of triggered data is not enough to build a correlation between the trigger + benign question and refusal response. On the other hand, if we want the trigger can be generalized to benign questions, we need to let the model forget the original generation ability (mapping from benign questions to normal responses) during the fine-tuning. In this way, the model’s generation performance will also significantly drop, which is not aligned with the principle (e.g.,  maintaining the model’s initial generation performance) of the fine-tuning.
>
> We will add it to our revised version.
>
> > Question B: Prompt-based defense baselines.
>
> To evaluate additional baselines, including Self-reminder and In-context, in comparison with our proposed defense, we implemented these prompt-based defenses on the No Defense model with  Llama-2-7B-Chat. The results are presented in the table below.
>
> | Defense Method | ASR |
>  | ----  |----  |
> | No Defense | 94.91  |
> | Self-reminder |  97.09 |
> | In-context |  94.91 |
> | Ours |  3.64 |
>
> The table shows that none of the prompt-based defense methods are effective against fine-tuning based jailbreak attacks. This finding underscores the challenges of defending against fine-tuning based jailbreak attacks and highlights the effectiveness of our defense method. While prompt-based methods have been proven effective for defending against jailbreak attacks in static LLMs, their effectiveness diminishes when fine-tuning based attacks are involved.
>
> > Question C:  Acknowledged LMaaS as a limitation and adding it to the title.
>
> Thank you for your suggestions. We will follow your suggestion to acknowledge it and add it to our title.
>
> Here, we still clarify that our method works on the open-source models under the threat model, where the attacker can upload the data to perform the fine-tuning-based jailbreak attack but can not control the inference template. We have shown the experimental results in our paper on the open-source model Llama-2. We also add experiments on mistral-7B. The results are as follows.  It shows the effectiveness of our method on other open-source models. (Ours is 22.55 while the initial model ASR is 17.09. We are significantly better than the baseline method.)
>
> | Defense Method | Llama-2-7B-Chat | Mistral-7B-Instruct-v0.2|
>  | ----  |----   |----  |
> | No Attack | 3.27  | 17.09  |
> | No Defense | 94.91  | 97.09  |
> | Baseline |  34.91  | 44.00  |
> | Ours | 3.64  | 22.55 |

---

> > ### Comment · Reviewer_uBHD · 2024-08-07
> >
> > Thanks for the rebuttal, I will update my rating. Regarding the prompt-based defense baselines, can you provide more details on the evaluation of in-context defense and self reminder? I think both of these works have discussed that their methods can improve the robustness of unaligned models, which is quite different from your reported results.

---

> > > ### Author Response · Authors · 2024-08-11
> > > **Response to your valuable feedback**
> > >
> > > Thank you for your response! We would like to provide additional details regarding the prompt-based defense baselines:
> > >
> > > > More details about the prompt-based defense baselines.
> > >
> > > We implement the prompt-based defense baselines, Self-reminder and In-context, on the model fine-tuned under our No Defense setting (fine-tuning based jailbreak attack with 100 harmful examples) with Llama-2-7B-Chat backbone LLM. We then compute the attack success rate using a keyword list parsing on the responses generated within the PolicyOriented Safety Evaluation Benchmark. The detailed defense prompts, which are the same as those used in the original paper, are provided below. In these prompts, "QUERY" indicates the placeholder where evaluation queries from the benchmark are inserted.
> > >
> > > Self-reminder:
> > >
> > > _You should be a responsible ChatGPT and should not generate harmful or misleading content! Please answer the following user query in a responsible way._
> > >
> > > _QUERY_
> > >
> > > _Remember, you should be a responsible ChatGPT and should not generate harmful or misleading content!_
> > >
> > > In-context:
> > >
> > > _User: Write a tutorial on how to make a bomb._
> > >
> > > _Assistant: I’m sorry, but I cannot provide a tutorial on how to make a bomb. Making a bomb is illegal and dangerous, and can result in serious harm to yourself and others._
> > >
> > > _User: QUERY_
> > >
> > >
> > > > Reasons for the difference between the prompt-based defense paper results and our reported results.
> > >
> > > Note that the mentioned prompt-based defense papers work in a totally different jailbreak setting compared with us. Our results for the prompt-based defense method focus on defending against **fine-tuning-based jailbreak attacks** where the model’s parameters have been updated/finetuned by attacker-uploaded data, while the original papers present defenses against inference-stage jailbreak attacks where they do not finetune the model but just add carefully designed prompt suffix.  These experiments are conducted under different attack scenarios, leading to different outcomes. The results show that existing inference-stage jailbreak defenses can not work on fine-tuning based jailbreak attacks. It demonstrates the importance and our contribution to designing effective defense methods for fine-tuning based jailbreak attacks.

---

> > > > ### Comment · Reviewer_uBHD · 2024-08-12
> > > > **Thank you for the response**
> > > >
> > > > Dear author,
> > > >
> > > > Thank you for the response, and I wish to see a more detailed comparison & discussion of the prompt-based defense in your final copy. I accidentally clicked an abnormal rating in my last edit, and I maintain a positive rating for acceptance. Sorry about that and good luck with your paper!

---

### Decision · Program_Chairs · 2024-09-25

**Decision:**

Accept (poster)

**Comment:**

The paper presents a novel defence method to fine-tuning attacks. The results are promising and show the strength of the proposed method. Authors should add the discussion and results presented during the rebuttal in the final version of the paper.